# Design of Hierarchically Tailored Hybrids Based on Nickle Nanocrystal-Decorated Manganese Dioxides for Enhanced Fire Safety of Epoxy Resin

**DOI:** 10.3390/ijms232213711

**Published:** 2022-11-08

**Authors:** Yao Yuan, Chen Liang, Anthony Chun Yin Yuen, Lulu Xu, Bin Yu, Chengfei Cao, Wei Wang

**Affiliations:** 1Fujian Provincial Key Laboratory of Functional Materials and Applications, School of Materials Science and Engineering, Xiamen University of Technology, Xiamen 361024, China; 2School of Mechanical and Manufacturing Engineering, University of New South Wales, Sydney, NSW 2052, Australia; 3School of Materials Science and Engineering, Nanyang Technological University, 50 Nanyang Avenue, Singapore 639798, Singapore; 4State Key Laboratory of Fire Science, University of Science and Technology of China, Hefei 230026, China; 5Centre for Future Materials, University of Southern Queensland, Springfield Central, Ipswich, QLD 4300, Australia

**Keywords:** polymer-matrix composites, surface treatments, flame retardancy, smoke toxicity

## Abstract

A novel and hierarchical hybrid composite (MnO_2_@CHS@SA@Ni) was synthesized utilizing manganese dioxide (MnO_2_) nanosheets as the core structure, self-assembly chitosan (CHS), sodium alginate (SA) and nickel species (Ni) as surface layers, and it was further incorporated into an epoxy matrix for achieving fire hazard suppression via surface self-assembly technology. Herein, the resultant hybrid epoxy composite possessed an exceptional nano-barrier and synergistic charring effect to aid the formation of a compact layered structure that enhanced its fire-resistive effectiveness. As a result, the addition of only 2 wt% MnO_2_@CHS@SA@Ni hybrids led to a dramatic reduction in the peak heat release rate and total heat release values (by ca. 33% and 27.8%) of the epoxy matrix. Notably, the peak smoke production rate and total smoke production values of EP/MnO_2_@CHS@SA@Ni 2% were decreased by ca. 16.9 and 38.4% compared to the corresponding data of pristine EP. This was accompanied by the suppression of toxic CO, NO release and the diffusion of thermal pyrolysis gases during combustion through TG-IR results. Overall, a significant fire-testing outcome of the proposed hierarchical structure was proven to be effective for epoxy composites in terms of flammability, smoke and toxicity reductions, optimizing their prospects in other polymeric materials in the respective fields.

## 1. Introduction

Epoxy resin (EP) as a popular option for the advanced manufacturing of coatings, flooring and paints has been widely used in various applications, including electronic insulation devices, semiconductor encapsulation, transportation and construction, on account of its exceptional properties, e.g., superior adhesive strength, high electrical insulation, easy implementation and low cost [1,2,3,4]. Nonetheless, like other organic polymer materials, EP possesses inherent flammability; it can be easily ignited and release a vast production of burning smoke and toxicity, which greatly restricts its further practical applications [5,6]. A considerable portion of previous efforts focused on the flame-retardant effectiveness of EP and relevant environmental concerns, e.g., the release of noxious or harmful smoke and gases, are usually neglected [7,8,9]. Accordingly, designing and developing new environment-friendly, flame-retardant alternative systems with exceptional smoke suppression properties is attractive and vital to environmental safety and community health.

Owing to the catalytic coupling effect, transition metal compounds have been extensively investigated with the purpose of removing the emissions of hazardous materials. The catalyst could offer additional chemical pathways for pyrolysis fragments to suppress the release of smoke and byproducts [10]. Xu et al. [11] incorporated copper and ferric oxides into rigid polyurethane foam for decreased pyrolysis toxic gases and smoke production rate. It is strongly believed that the smoke and gas toxicity were effectively suppressed with the incorporation of transition metal-based flame retardant. As one of the transition metal oxides, manganese dioxides (MnO_2_) have attracted considerable attention in view of their natural abundance, cost-effectiveness, ease of preparation and environmental compatibility [12]. Further, multiple morphologies of MnO_2_ nanomaterials such as rods, sheets, flowers, belts and dendrites can be obtained using various synthetic approaches.

Among the morphological structures of flame-retardant materials, two-dimensional materials, e.g., MXene [13], molybdenum disulfide [14] and graphene [15,16], as well as their inorganic–organic multifunctional hybrids [17], have received major research attention. A previous research work revealed that inorganic materials with layered structures have unique advantages compared to those with a one-dimensional and three-dimensional structure [18]. Moreover, MnO_2_ nanosheets have demonstrated their flame-retardant effectiveness in a polymer matrix during combustion. However, the introduction of untreated MnO_2_ was previously found not to be sufficient as the core FR additive to effectively improve flame retardancy for EP [19].

Surface modification of inorganic material is the key strategy affecting the structure and property of polymer composites, since it can significantly enhance the functionality of nanofiller-based flame retardants. Geng and co-authors [20] grafted the negatively charged surfaces of MnO_2_ nanosheets with aniline monomer by electrostatic interactions and situ polymerization, producing a three-dimensional macroscopic network, which possessed an excellent capacitance and cycling performance. Simultaneously, surface functionalization of inorganic material with flame retardants has been proved to affect the dispersibility and interfacial adhesion between polymer and additive fillers [6], which can satisfy specific requirements, such as higher thermal stability and excellent smoke suppression capability. Notably, fabricating the interfacial metal species-supported hybrid is of importance for fire safety design. Hence, it is of great significance to modify MnO_2_ nanosheets with eco-friendly hybrid organic–inorganic materials, which possess a high interfacial effect and provide better catalytic efficiency.

Considering the requirement of environment protection, biomass materials, e.g., chitosan and alginate, are considered to be desirable modifier candidates to functionalize MnO_2_ nanosheets. In fact, owing to their widespread accessibility, low price, compatibility and non-toxicity, chitosan and alginate have been used as effective supports in the functionalization and fabrication of metal oxide complexes. Moreover, it is noteworthy that related research has indicated that the chitosan- and alginate-based hybrids can serve as promising green charring agents in flame-retardant composites [21].

Aiming to achieve fire hazard suppression of EP composites, a three-layer hierarchical hybrid based on a MnO_2_ nanosheet decorated with chitosan (CHS), sodium alginate (SA) layers and nickel species was constructed via surface self-assembly technology. As shown in Appendix A, the CHS and SA were employed to act as positively and negatively charged polyelectrolytes towards the fabrication of eco-friendly and high-performance epoxy composites. Additionally, the nickel species have a strong complexation ability to SA and thus enhance the strength of the chars during the combustion process of EP composites [22]. Validated by a series of structure and morphology characterizations, the introduction of 2% MnO_2_@CHS@SA@Ni hybrids into EP can achieve fire and smoke suppression. Through systematical investigation of the catalytic charring behavior in the condensed phase and gaseous phase by various measurements, the tailored three-layer hierarchical hybrids enable the enhancement of the catalytic charring of EP/MnO_2_@CHS@SA@Ni composites, which is beneficial for forming a strong physical nano-barrier and increasing the fire safety of the epoxy matrix. Accordingly, the self-assembled, hierarchically tailored hybrids, meanwhile, provide an innovative path for the preparation of highly thermal, stable, excellent fire-resistant and superior smoke-suppressive epoxy composites.

## 2. Results

### 2.1. Characterization of MnO_2_ and MnO_2_@CHS@SA@Ni Hybrids

The synthetic route of the target product MnO_2_@CHS@SA@Ni hybrids with a three-layer structure is depicted in Figure 1a. As shown, the bio-based CHS and SA layers were generated onto the external surface of MnO_2_ nanosheets due to the electrostatic interactions between the three components. After the incorporation of Ni^2+^ ions, the nickel nanoparticles were anchored with alginate and fixed on the surface, resulting in the target product, which was defined as MnO_2_@CHS@SA@Ni hybrids.

To unveil the morphology and structure of the MnO_2_@CHS@SA@Ni hierarchical hybrids, a series of measurements was performed. As depicted in Figure 1b–d, untreated MnO_2_ exhibits a lamellar structure with enormous wrinkles and inhomogeneous dimensions of several hundred nanometers. Figure 1e–h presents the typical TEM images of MnO_2_@CHS@SA@Ni hybrids with a multilayer and wrinkled structure, showing a uniform dispersion of ultrafine nickel species on the surface of the MnO_2_ hierarchical hybrids. Additionally, it can be observed in Figure 1h that the black dots are evenly distributed on the surface of MnO_2_ nanosheets, with a diameter of around 4 nm. The HRTEM imaging utilized a lattice spacing of 0.21 nm and 0.20 nm for the black dots, which are, respectively, attributed to the (111) plane of fcc-Ni and supposed to be Ni nanocrystals. Furthermore, the interplanar spacing of 0.70 nm is derived from the (001) lattice plane of δ-MnO_2_ [23]. As depicted in Figure 1j, the characteristic peaks of the Mn, Ni, O, N and C elements from the MnO_2_ nanosheets, chitosan, alginate and nickel are confirmed by EDX mapping of MnO_2_@CHS@SA@Ni hybrids. As also evidenced in Figure 2a–h, the TEM images and related element mapping results indicated that the MnO_2_@CHS@SA@Ni hybrid was prepared successfully, and a good dispersion of nickel dots on the surface of the MnO_2_ nanosheets was successfully realized.

X-ray photoelectron spectroscopy (XPS) was employed to analyze the surface chemical structure of untreated MnO_2_ and MnO_2_@CHS@SA@Ni hybrids (Appendix A). In the case of the MnO_2_@CHS@SA@Ni hybrids, the characteristic peaks of N and Ni elements could be observed, which are derived from the chitosan and nickel salt. As depicted in Appendix A, the Mn2p signals of the untreated MnO_2_ and MnO_2_@CHS@SA@Ni hybrids are divided into the locations at 642.8 eV and 654.6 eV, corresponding to the Mn 2p3/2 and Mn 2p1/2 binding energies, showing the Mn (IV) state in both untreated MnO_2_ and MnO_2_@CHS@SA@Ni hybrids.

The fitting of the N1s spectra of the MnO_2_@CHS@SA@Ni hybrids (Figure 2i) discloses peaks at 399.8 eV and 401.9 eV, respectively, which are assigned to the primary amines from chitosan with and without protonation [24]. Similarly, Figure 2j exhibits the high-resolution Ni 2p spectra. The characteristic peak at 856.2 eV for Ni 2p3/2 is due to metallic Ni in the hierarchical structure. In essence, the corresponding satellite peaks of Ni 2p3/2 and Ni 2p1/2 are situated at 861.0 eV and 879.8 eV, respectively [25]. As shown in Figure 2l, the binding energies in the high-resolution C1s spectra of the MnO_2_@CHS@SA@Ni hybrids are located at 284.8 eV, 286.6 eV and 288.2 eV, which are attributed to the C-C/C-H, C-O/C-OH and COOH, showing the successful coating of CHS and SA layers.

The chemical structure of the MnO_2_@CHS@SA@Ni hybrids was verified utilizing FTIR spectroscopy, as shown in Appendix A. In general, the absorption at ~3420 cm^−1^ is ascribed to the vibrations of H-bonded O-H and N-H groups. For the MnO_2_@CHS@SA@Ni hybrids, the peak that appeared at 1047 cm^−1^ corresponds to the C-N stretching vibration, which verifies the inner layer of CHS [26]. Further, the presence of absorption at 1597 and 1420 cm^−1^ is ascribed to the -COO- stretching vibration, which indicates that the SA layer is anchored on the surface of MnO_2_ hierarchical hybrids. Moreover, the absorption at 578 cm^−1^ contributes to the vibration of the Ni-O bond. The thermal decomposition of untreated MnO_2_ and MnO_2_@CHS@SA@Ni hybrids was analyzed by thermogravimetric analysis. A slight weight loss is observed for untreated MnO_2_ under anaerobe conditions (~11.5% mass loss), as showed in Appendix A. By comparison, the higher weight loss of the MnO_2_@CHS@SA@Ni hybrids (about 56.9% mass loss) can be discovered. It could be ascribed to the decomposition of organic CHS and SA, which is in agreement with the TEM, HRTEM images, XPS spectra and FTIR analyses.

### 2.2. Thermal Properties of Pristine EP and Its Composites

Thermogravimetric analysis (TGA) was carried out to investigate the thermal degradation behaviors of samples under a nitrogen atmosphere (Figure 3a and Appendix A), and the relevant data are provided in Appendix A. The EP composites display a single-step thermal decomposition process and a similar degradation process. When 2 wt% MnO_2_ is applied, the T_5%_ (temperature at 5 wt% weight loss) and T_max_ (temperature at maximum mass loss rate) of EP/MnO_2_ composite are reduced by 19 °C and 8 °C, respectively. However, after the incorporation of 2 wt% of MnO_2_@CHS@SA@Ni hybrids, distinctly decreased T_5%_ and T_max_ are observed compared with EP/MnO_2_ composite, which is ascribed to the unstable organic components of the outer layers. Simultaneously, as the MnO_2_@CHS@SA@Ni content increases from 0.5 to 2 wt%, the char yield of the EP/MnO_2_@CHS@SA@Ni composite is improved by 43.9% compared with pristine EP. In Figure 3b, the DTG curves indicate that the maximum mass-loss rates of EP/MnO_2_@CHS@SA@Ni composites are exceedingly smaller than that of pristine EP. Accordingly, the MnO_2_@CHS@SA@Ni hybrids could facilitate the char formation of EP and restrict the mass/energy transfer, implying the suppression of the release of volatiles and improvement of fire safety by catalytic charring behavior [27].

### 2.3. Fire safety Analysis of Pristine EP and Its Composites

Benchmark fire tests of the EP composites were carried out by cone calorimeter, and the results together with key figures are provided in Figure 3c–f and Table 1. It is worth noting that pristine EP is easily ignitable and reaches the first peak at 125 s, exhibiting incisive heat release rate (HRR) curves, with a peak of heat release rate (PHRR) of 1657 kW/m^2^ and total heat release rate (THR) of 79.8 MJ/m^2^. By varying the MnO_2_@CHS@SA@Ni contents from 0.5 to 2 wt%, compared to that of pristine EP, the PHRR values are reduced directly by 14.6%, 28.6% and 33%, and THR values decrease to 66.4, 63.9 and 57.6 MJ/m^2^, respectively. As a control experiment, by adding 2 wt% of untreated MnO_2_ and MnO_2_@CHS@SA@Ni hybrids, a decrease of 20.1 and 27.8% in THR compared to pristine EP is acquired, corresponding to the physical nano-barrier and catalytic charring effect of the MnO_2_@CHS@SA@Ni hybrids. These results certify that the incorporation of MnO_2_@CHS@SA@Ni hybrids into EP exhibits a significant effect on decreasing the PHRR and THR in combustion and achieve great improvement in the flame retardancy of EP/MnO_2_@CHS@SA@Ni composites [28].

Generally, epoxy resin can be ignited easily and rapidly produces a large number of toxic gases (e.g., carbon monoxide/dioxide, hydrogen cyanide) and visible smoke particles that could cause human death in a short time and increase the difficulty of rescue. Figure 3e,f present the curves of smoke production rate (SPR) and total smoke production (TSP) versus time for the EP composites. Typically, pristine EP shows the peak SPR value of 0.59 m^2^/s and TSP value of 27.36 m^2^. Unfortunately, the peak SPR of EP/MnO_2_ gives a higher value, indicating the introduction of untreated MnO_2_ failed to effectively reduce the toxic smoke of the EP matrix during combustion. In terms of 2 wt% MnO_2_@CHS@SA@Ni hybrids-based EP composites, the peak SPR and TSP values decrease by 16.9% and 39.0%, respectively, revealing the property of improved smoke suppression.

To further evaluate the fire safety, the fire growth rate (FIGRA) and the fire performance index (FPI) are plotted in Appendix A [29], and the formulas are expressed as follows:(1)$$\mathrm{FIGRA}=\frac{\mathrm{PHRR}}{\mathrm{tPHRR}}$$
(2)$$\mathrm{FPI}=\frac{\mathrm{TTI}}{\mathrm{PHRR}}$$

Based on Table 1 and the formulas, pristine EP shows a FIGRA of 13.24 kW/m^2^·s and FPI of 0.037 m^2^·s/kW. Generally, the decreased FPI values represent the premature flash over and thus the increased FPI values are satisfying for fire safe EP composites. Especially, the FIGRA of EP/MnO_2_ and EP/MnO_2_@CHS@SA@Ni with 2 wt% fillers are reduced by 22.2% (10.29 kW/m^2^·s) and 31.4% (9.08 kW/m^2^·s), manifesting a shorter time to PHRR. Conversely, the increased FPI value is satisfying and indicates improved fire safety. When the MnO_2_@CHS@SA@Ni content increases from 0.5 to 2 wt%, the FPI values are enhanced directly by 10.8%, 35.1% and 56.8%. Accordingly, EP/MnO_2_@CHS@SA@Ni 2% exhibits the highest FGI and FPI values, demonstrating that this bio-based flame-retardant hybrid has a significant effect on enhancing the fire safety.

### 2.4. Gaseous Phase Analysis

Successively, to detect the evolved volatiles of pristine EP and its composites during thermal decomposition and evaluate the toxic smoke products of EP composites, TGA-FTIR (TG-IR) measurement is employed. Figure 4a presents the three-dimensional FTIR spectra of pyrolysis gases, which are at maximum decomposition rates during degradation. The results show that the obtained samples reach the maximum decomposition rates near 480 °C and possess similar spectra. For pristine EP shown in Figure 4b, a strong absorption signal is observed compared with EP/MnO_2_ and EP/MnO_2_@CHS@SA@Ni. Further, the emergence of the peaks at 3656 cm^−1^, 3034 cm^−1^, 2930 cm^−1^, 2360 cm^−1^, 1738 cm^−1^, 1510 cm^−1^ and 1174 cm^−1^ can be clearly observed, which are respectively derived from -OH, -CH_2_-, hydrocarbons, CO_2_, carbonyl compounds, aromatic compounds and esters [14].

To further make direct comparison of the evolved volatiles, the relative intensities of major pyrolysis products of EP, EP/MnO_2_ and EP/MnO_2_@CHS@SA@Ni are portrayed in Figure 5. From the Gram–Schmidt curve, it can be seen that the absorbance intensities of released volatiles are dramatically decreased with the addition of MnO_2_ and MnO_2_@CHS@SA@Ni hybrids. Owing to the physical adsorption of MnO_2_ nanosheets and the deposition of bio-layers loading on the surface of MnO_2_ nanosheets, EP/MnO_2_@CHS@SA@Ni presents the lowest intensity, and the signal intensity of the pyrolytic products, e.g., hydrocarbons, aromatic compounds and esters, have been remarkably decreased, and they can act as additional fuels during combustion and aggregate to form toxic smoke. Meanwhile, the release of toxic CO and NO (i.e., asphyxiated gases that cause human suffocation and respiratory disease) is significantly decreased, which is ascribed to the exceptional nano-barrier and catalytic charring behavior of the hierarchical hybrids composite, promoting the formation of a compact layered structure that effectively suppresses fire toxicity.

### 2.5. Condensed Phase Analysis

The physical structure of the residual chars after the cone calorimeter test is employed to explore the condensed phase strategy associated with the flame-retardant mechanism. The digital photos of residual chars are presented in Figure 6a–d. Usually, it is widely acknowledged that pristine EP burns severely, leaving cracked and brittle residual chars. For EP/MnO_2_@CHS@SA@Ni 2%, it exhibits the highest residual chars with a certain amount of intumescent phenomenon. Figure 6e–h shows that the pristine EP has obvious cracks and fissures after combustion, thereby leading to the permeation of heat and pyrolysis products and incapability to protect the EP matrix from further degradation. For EP/MnO_2_@CHS@SA@Ni 2%, the char layer becomes smooth and intact, with no collapse and cracks, which acts as an effective “physical barrier” that prevents heat from penetrating through while lowering the migration of combustible gases on the material surface.

Apart from the morphologies, the upper layers of the condensed-phased products after cone calorimeter tests were also collected to investigate the chemical structure and degree of graphitization by Raman spectroscopy shown in Figure 6i–l. As can be observed, all the samples present two visible bands, namely, the D-band and G-band located at ca. 1350 cm^−1^ and 1593 cm^−1^, respectively. Usually, the ratio of the intensity of the D-band and G-band (I_D_/I_G_) is used to evaluate the graphitization degree of the char residues [30,31]. A lower value of I_D_/I_G_ means a better graphitic structure and physical insulation. The results indicate that the introduction of MnO_2_@CHS@SA@Ni hybrids can significantly reduce the I_D_/I_G_ value, confirming that the as-prepared bio-based hybrids can improve the graphite layers’ formation of the EP matrix and protect underlying material during the degradation process.

The XPS technique is employed to confirm the chemical structure and composition of the residual chars. Figure 6m–p shows the high-resolution C1s spectra of interior and exterior chars for pristine EP and EP/MnO_2_@CHS@SA@Ni after cone calorimeter test. As depicted, the C1s spectra can be deconvoluted to three peaks situated at 284.6 eV, 285.6 eV and 288.3 eV, which are, respectively, ascribed to C-H/C-C (aromatic and aliphatic species), C-O and C=O (hydroxyl and ether group). In essence, C_ox_ means oxidized carbon; Ca means aliphatic and aromatic carbons, and the lower value of C_ox_/C_a_ is commonly employed to demonstrate the thermal oxidative resistance of the EP matrix [18]. For the interior char, the value of C_ox_/C_a_ of EP/MnO_2_@CHS@SA@Ni (0.23) is lower than that of pristine EP (0.79). In the case of exterior char, the value of C_ox_/C_a_ of EP/MnO_2_@CHS@SA@Ni is 0.26, much lower than that of pristine EP (0.80), demonstrating a significant improvement. Furthermore, the XPS survey spectra and (b,c) elemental composition for pristine EP and EP/MnO_2_@CHS@SA@Ni 2% are summarized in Figure 7, as determined by XPS. It is well-known that the peak at 641.3 eV is derived from Mn 2p, and the signal at 855.6 eV corresponds to Ni 2p, which may be expected to exhibit enhanced catalytic activity for reducing the fire hazards during decomposition. Hence, it can be concluded that the thermal oxidative resistance of EP/MnO_2_@CHS@SA@Ni composites exhibits better effect than that of pristine EP, which is consistent with the Raman spectrum results.

### 2.6. Potential Flame-Retardant Mechanism

In terms of these above-mentioned results and discussions, the flame-retardant mechanism for the enhanced fire safety of EP composites is proposed in Figure 7. In the aspect of the typical lamellar structure, MnO_2_ nanosheets can play the role of physical barriers to restrict the movement of polymer chains and the evolution of decomposed products. Meanwhile, the CHS and SA layers act as efficient flame-retardant charring agents in the epoxy, which promote the formation of protective chars. Additionally, the presence of Ni species anchored by alginate on the surface of MnO_2_ nanosheets with catalytic activity is conducive to a reduction in smoke toxicity for EP. Specifically, after the incorporation of the three-layer bio-based flame-retardant hybrids, useful barriers with graphitized chars were generated during the combustion, which can hinder the decomposed volatiles, heat and oxygen, and thus protect underlying material.

## 3. Materials and Methods

### 3.1. Materials

Chitosan (CHS) (molecular weight 74.1 kDa) with a 96.56% degree of deacetylation and sodium alginate (SA) (viscosity: 200 ± 20 mPa.s, Mw = 398.31 kDa, mannuronate/guluronate ratio = 1.1) were obtained from Aladdin Reagents Shanghai Co., Ltd. (Shanghai, China). Potassium permanganate (KMnO_4_), nickel chloride (NiCl_2_·6H_2_O), manganese chloride (MnCl_2_·4H_2_O), ethyl acetate and diaminodiphenylmethane (DDM) were all provided from Sinopharm Chemical Reagent Co., Ltd. (Shanghai, China). Bisphenol-A type EP (commercial code: E-44, EEW: 227 g/equivalent) was supplied by Shixian Chemical Industry Co., Ltd. (Shenyang, China).

### 3.2. Synthesis of MnO_2_ Nanosheets and MnO_2_@CHS@SA@Ni Hybrids

MnO_2_ nanosheets were synthesized through the wet-chemical method. An amount of 15 mmol of KMnO_4_ was added into a three-necked bottle with 750 mL of deionized water. Until the KMnO_4_ was completely dissolved in water, 200 mL of ethyl acetate was added into the deep-purple solution by constantly stirring. Afterwards, the above solution was maintained at 95 °C and refluxed overnight. The brown products were filtrated, washed consecutively and dried at 70 °C overnight. In the case of MnO_2_@CHS@SA@Ni hybrids, surface self-assembly technology was employed. Firstly, 1 g of MnO_2_ powder was dispersed in 500 mL deionized water with continuous agitation. Then, 100 mL of positive charged CHS solution with a concentration of 0.2 wt% (pH = 5) was added dropwise into the above suspension. After constantly stirring for 2 h, the suspension was centrifuged and washed up to three times. Afterwards, the resultant hybrid was redispersed in 500 mL of water under stirring and ultrasonic. The 100 mL of prepared SA solution with a 0.3 wt% concentration was dropped into the suspension. After 2 h stirring, the suspension was centrifuged and washed and rewashed up to three times. The sediment was also redispersed in 500 mL of water. Lastly, 10 mL of nickel chloride aqueous solution (30 mmol/L) was added dropwise into the mixture. After being maintained for 2 h, the suspension was centrifuged and washed in water and dried at 80 °C overnight.

### 3.3. Preparation of Pristine EP and Its Composites

An amount of 1 g of MnO_2_@CHS@SA@Ni was first added into 40 mL acetone by ultrasonication and stirring. After 2 h, 40.83 g of the pre-melting epoxy resin was poured into the suspension with continuous agitation. After that, the above suspension was maintained at 110 °C for 5 h to remove the solvent. Then, 8.17 g of pre-melting DDM was added into the mixture for 2 min and transferred to the model. Lastly, the EP composite was heated at 100 °C and 150 °C for 2 h, and the final sample was remarked as EP/MnO_2_@CHS@SA@Ni 2%. Other samples were prepared using similar procedures.

### 3.4. Characterization

Transmission Electron Microscopy Tests. Transmission electron microscopy (TEM) images were obtained using a Jeol JEM-100SX transmission electron microscope with an acceleration voltage of 100 kV.

Thermogravimetric Analysis (TGA). TGA of samples was undertaken using a TGA-Q5000 apparatus (TA Co., Boston, MA, USA) from 50 °C to 700 °C at a heating rate of 20 °C min^−1^. The weight of all samples was maintained within 3–5 mg in an open platinum pan.

Fourier Transform Infrared Spectroscopy. Fourier transform infrared spectroscopy (Nicolet 6700 FT-IR spectrophotometer, Thermo Fisher Scientific, Waltham, MA, USA) was employed to characterize MnO_2_ hybrids using a thin KBr disc. The transmission mode was used, and the wavenumber range was set from 4000 to 400 cm^−1^.

Scanning Electron Microscopy (SEM) Test. The morphologies of MnO_2_ materials with three different dimensions, coated with a gold layer in advance, were observed using scanning electron microscopy (SEM; AMRAY1000B, Beijing R&D Center of the Chinese Academy of Sciences, Beijing, China).

Thermogravimetric Analysis-Fourier Transform Infrared Spectrometry (TGA-FTIR). TGA-FTIR of the samples was performed using a TGA Q5000IR thermal gravimetric analyzer that was interfaced with the Nicolet 6700 FTIR spectrophotometer. Approximately 5.0 mg of the sample was placed in an alumina crucible and heated from 30 °C to 800 °C at a heating rate of 20 °C min^−1^ (helium atmosphere, flow rate of 45 mL min^−1^).

Cone Calorimeter Test. A combustion test was performed on the cone calorimeter (Fire Testing Technology, East Grinstead, UK.) according to ISO 5660 standard procedures, using 100 × 100 × 3 mm^3^ specimens. Each specimen was exposed horizontally to 35 kW m^−2^ external heat flux.

X-ray photoelectron spectroscopy (XPS). XPS was performed on a VG Escalab Mark II spectrometer (Thermo-VG Scientific Ltd., Waltham, MA, USA) using Al Kα excitation radiation (hυ = 1486.6 eV).

Raman spectroscopy. Raman spectra of the char after cone calorimeter tests were recorded on a SPEX-1403 laser Raman spectrometer (SPEX Co., Metuchen, NJ, USA)

## 4. Conclusions

In this work, the hierarchical assembly of Ni nanocrystals on MnO_2_ nanosheets interconnected with bio-based chitosan (CHS) and sodium alginate (SA) was thoughtfully designed and achieved via a facile wet chemical method. This tailored three-layer hierarchical hybrid (MnO_2_@CHS@SA@Ni) structure was clearly observed and confirmed by TEM, HRTEM, EDX Mapping, XPS and FTIR. It was then incorporated into the EP matrix and resulted in a facile, effective flame-retardant composite with outstanding fire- and toxicity-suppression performances. With only 2 wt% MnO_2_@CHS@SA@Ni hybrids added, it was found that the PHRR and THR values were reduced by 33% and 27.8% compared to the pristine epoxy. Additionally, the peak SPR and TSP values of EP/MnO_2_@CHS@SA@Ni were decreased by 16.9 and 38.4% compared to the corresponding data of pristine EP, indicating significant reductions in epoxy smoke particulates owing to the rationally designed structure. The underlying flame-retardant mechanism that led to the improvement of the hybrid EP composite was discussed comprehensively based on the above-mentioned results. Specifically, the higher yields of graphitized char layers were confirmed by both burnt surface zone observation and chemical analysis. Moreover, the emission of toxic volatiles of EP/MnO_2_@CHS@SA@Ni was investigated by utilizing TG-IR. It was found that the introduction of MnO_2_@CHS@SA@Ni hybrids inhibited the release of CO, NO and other evolved volatiles, as well as limiting the smoke generation of the EP polymer matrix by eliminating the production of chemical byproducts and soot particle precursors. In summary, the three-layer architecture endows epoxy with remarkable enhancements in fire and smoke suppression. This design of surface-functionalized bio-based MnO_2_@CHS@SA@Ni hybrids with hierarchical structures containing organic and inorganic layers demonstrates multiple functionalities that overcome the weaknesses of epoxy resins. This developed hybrid EP composite structure could potentially expand the versatility of applications.

## Figures and Tables

**Figure 1 ijms-23-13711-f001:**
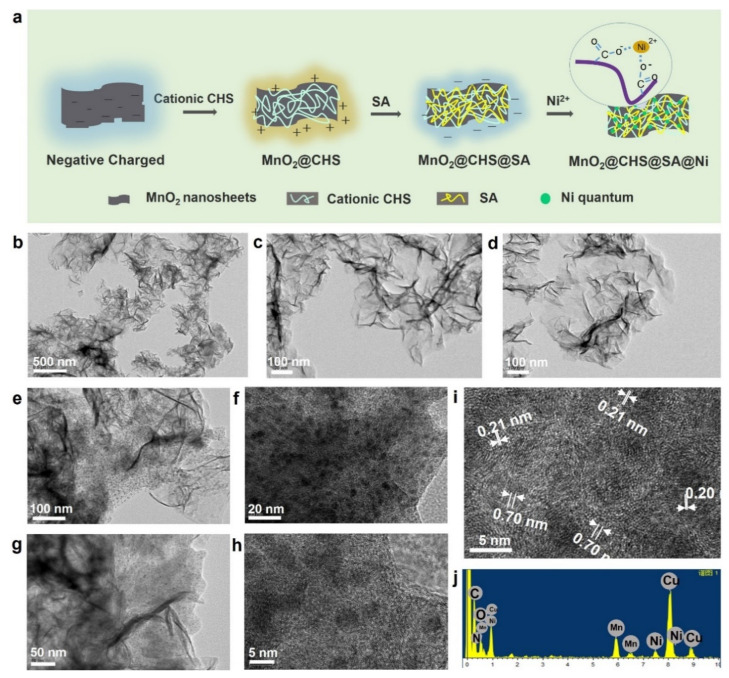
Schematic image of fabrication process and structure characterization of MnO_2_@Chitosan@SA@Ni hybrids. (**a**) Preparation scheme of MnO_2_@Chitosan@SA@Ni hybrids, (**b**–**d**) TEM image of untreated MnO_2_, (**e**–**h**) TEM, (**i**) HRTEM image and (**j**) EDX spectrum of MnO_2_@Chitosan@SA@Ni hybrids.

**Figure 2 ijms-23-13711-f002:**
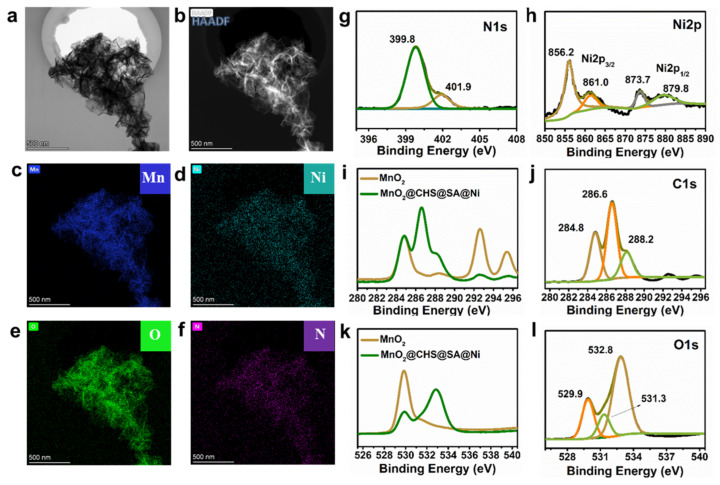
Structural analysis of MnO_2_@CHS@SA@Ni hybrids using TEM and XPS techniques. (**a**) TEM image, (**b**) HAADF image, (**c**–**f**) elemental mapping (Mn, Ni, O and N, scale bar: 500 nm) of MnO_2_@CHS@SA@Ni hybrids and the high-resolution (**g**) N1s, (**h**) Ni2p, (**j**) C1s, (**l**) O1s spectra of MnO_2_@CHS@SA@Ni hybrids, (**i**) C1s and (**k**) O1s spectra of untreated MnO_2_ and MnO_2_@CHS@SA@Ni hybrids.

**Figure 3 ijms-23-13711-f003:**
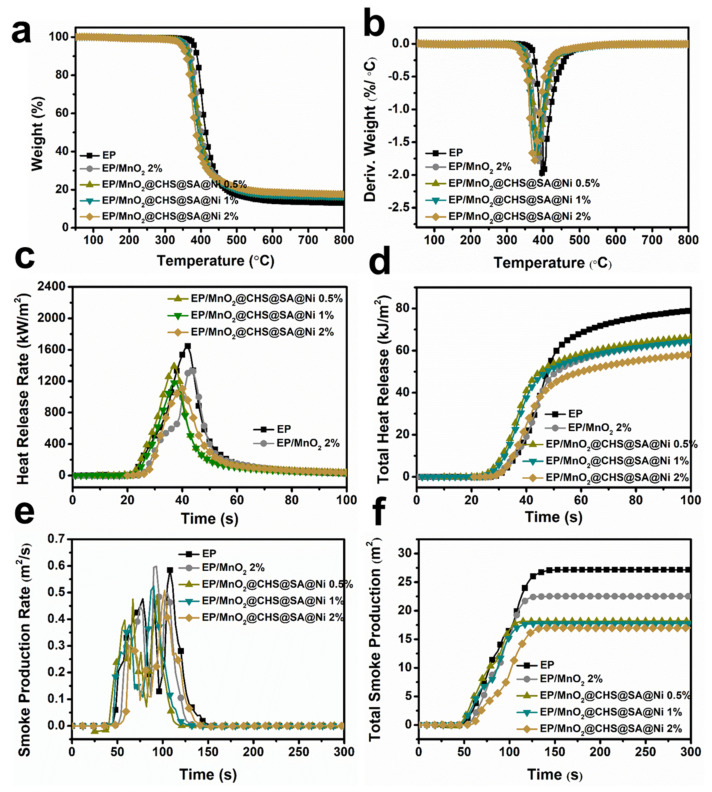
Thermal stability and fire resistance of control epoxy and its composites. (**a**) TGA, (**b**) DTG curves under nitrogen atmosphere, (**c**) heat release rate, (**d**) total heat release, (**e**) smoke production rate and (**f**) total smoke production versus time curves of pristine EP, EP/MnO_2_ and EP/MnO_2_@CHS@SA@Ni composites.

**Figure 4 ijms-23-13711-f004:**
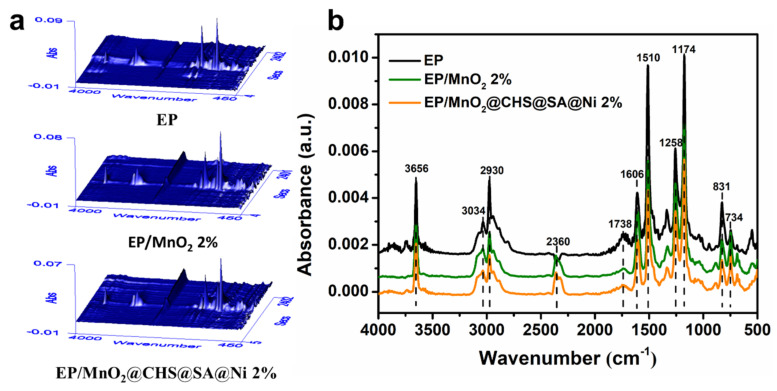
3D FTIR spectra of pyrolysis products at maximum decomposition rates during the degradation process. (**a**) 3D FTIR spectra of control EP, EP/MnO_2_ 2%, and EP/MnO_2_@CHS@SA@Ni 2%. (**b**) FTIR spectra of pyrolysis products at maximum weight loss rate of pristine EP, EP/MnO_2_ 2% and EP/MnO_2_@CHS@SA@Ni 2%.

**Figure 5 ijms-23-13711-f005:**
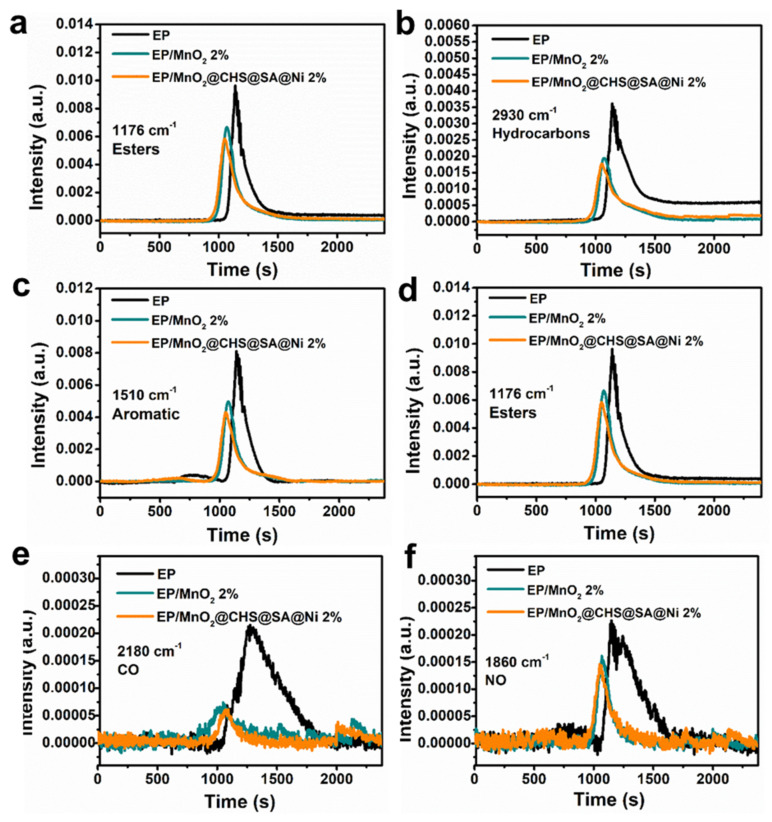
Absorbance spectra of pyrolysis products of pristine EP, EP/MnO_2_ 2% and EP/MnO_2_@CHS@SA@Ni 2% versus time: (**a**) Gram–Schmidt curves, (**b**) hydrocarbons, (**c**) aromatic compounds, (**d**) esters, (**e**) CO and (**f**) NO.

**Figure 6 ijms-23-13711-f006:**
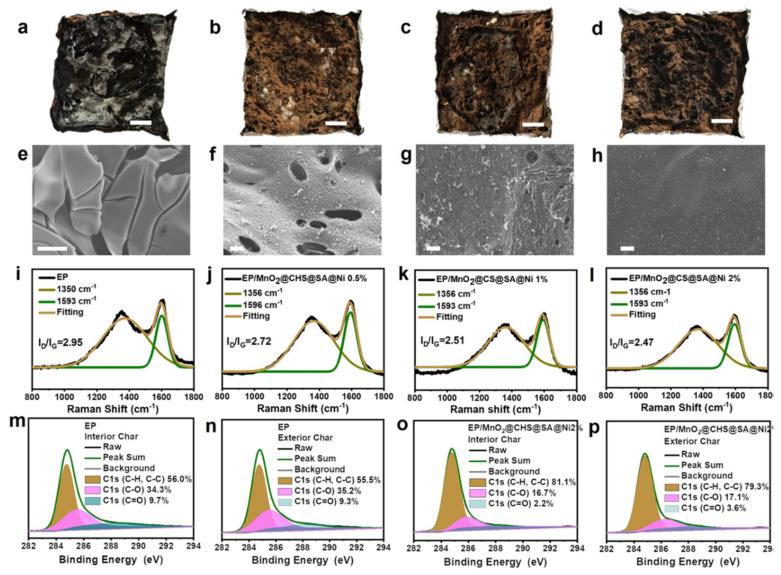
Char residue analysis of epoxy composites. (**a**–**d**) Digital images (the insert scale bar is 2 cm), (**e**–**h**) SEM images (the insert scale bar is 40 μm) and (**i**–**l**) Raman results of char residues for pristine EP, EP/MnO_2_@CHS@SA@Ni 0.5%, EP/MnO_2_@CHS@SA@Ni 1% and EP/MnO_2_@CHS@SA@Ni 2% and (**m**–**p**) the high-resolution C1s spectra of interior and exterior char residues for pristine EP and EP/MnO_2_@CHS@SA@Ni 2% after cone calorimeter test.

**Figure 7 ijms-23-13711-f007:**
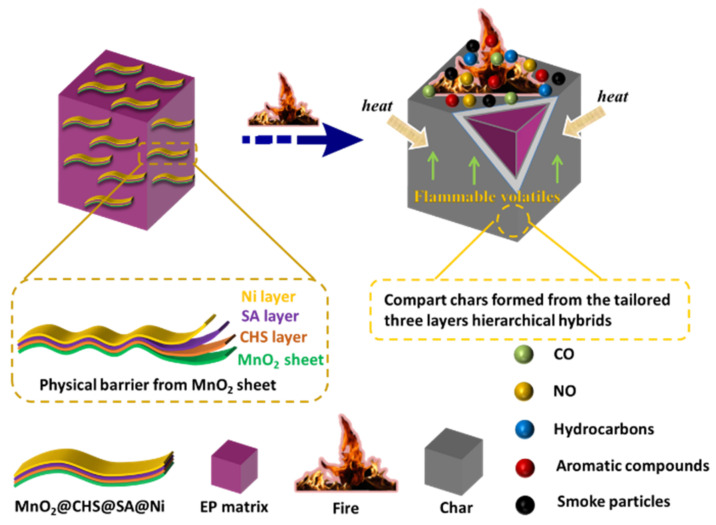
Potential flame-retardant mechanism of EP/MnO_2_@CHS@SA@Ni composites.

**Table 1 ijms-23-13711-t001:** Cone calorimeter test results of pristine EP and its composites.

Samples	TTI (s)	PHRR (kW/m^2^)	THR (MJ/m^2^)	FIGRA (kW/m^2^·s)	FPI (m^2^·s/kW)	TSP (m^2^)
EP	62	1655	79.2	13.24	0.037	27.1
EP/MnO_2_ 2%	64	1348	64.8	10.29	0.047	22.5
EP/MnO_2_@CHS@SA@Ni 0.5%	58	1426	66.1	12.84	0.041	18.4
EP/MnO_2_@CHS@SA@Ni 1%	59	1186	64.3	10.68	0.050	17.9
EP/MnO_2_@CHS@SA@Ni 2%	63	1090	57.7	9.08	0.058	16.7

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
