# Peer review of "Design of Hierarchically Tailored Hybrids Based on Nickle Nanocrystal-Decorated Manganese Dioxides for Enhanced Fire Safety of Epoxy Resin"

_ijms, 2022, doi:10.3390/ijms232213711_

Round 1
Reviewer 1 Report
Manuscript entitled “Design of Hierarchically Tailored Hybrids based on Nickle Nanocrystals Decorated Manganese Dioxides for Enhanced Fire Safety of Epoxy Resin” by Wang et. al., has introduced hierarchical bio-based flame retardant hybrids and application to epoxy resin composites. The manuscript is well written, the characteristics of the new bio-based materials, especially the fire and toxicity suppression performances, were investigated by using different techniques and the results were correctly interpreted. The overall readership can fulfill “International Journal of Molecular Sciences”. I think this manuscript can be acceptable after minor grammatical mistakes.
(1) Abstract. The “Ni” abbreviation did not have full name at the first appearance. The “nickel species” should be “nickel species (Ni)”. Please make sure that every abbreviation has its full name when it is described the first time.
(2) It is stated in the Thermogravimetric analysis section that the T-5% and Tmax of EP/MnO2 composite are decreased by 19 ℃ and 8 ℃, respectively. How do they define the T-5% and Tmax?
(3) The authors should unify the name of TGA measurement. Somewhere it is called TGA-FTIR, somewhere it is called TG-IR. This will make readers confusing.
(4) The char residue analysis of epoxy composites lost the scale bar in Figure 6. Specify 'a-h'.
(5) Line 93. “a three-layer hierarchical hybrid based on MnO2 nanosheets decorated with chitosan (CHS), sodium alginate (SA) layers and nickel species was constructed via surface self-assembly technology”. “MnO2 nanosheets” should be “MnO2 nanosheet”.
(6) Line 228. “Generally, the decreased FPI values represent the premature flash over and thus the in-creased FPI values are satisfying for fire safety EP composites.” The “for fire safety EP composites” should be either “for fire safe EP composites” or “for fire safety of EP composites”.
Reviewer 2 Report
The manuscript entitled “Design of Hierarchically Tailored Hybrids based on Nickle Nanocrystals Decorated Manganese Dioxides for Enhanced Fire Safety of Epoxy Resin” by Y. Yuan, C. Liang, L. Xu, B. Yu, C. Cao and W. Wang.
The manuscript presents a method for improving the fire-resistant of epoxy resins, using a hierarchical hybrid composite (MnO2@CHS@SA@Ni) as flame retardant. The novelty of the present manuscript consists in the preparation and characterization of surface-functionalized bio-based hybrids containing organic and inorganic layers. The introduction of the hybrids in epoxy resin was found to have a significant effect in improving the fire resistance and smoke suppression of the resulting composites.
Some remarks:
The description of Chitosan (CHS) should be given (molecular weight and degree of deacetylation). The description of sodium alginate (SA) should be given (mannuronate/guluronate ratio). It will be better if the author could add more similar published articles to show the effect of this kind of composites in this manuscript. There are a few misspelling as well: (“the increased FPI values are satisfying for fire safety EP composites.” should be “the increased FPI values are satisfying for fire safe EP composites” (line 228). “3.1. materials” should be “3.1. Materials”.
The manuscript is well written, the characteristics of the new composites, especially the smoke suppression and fire-resistant properties, were investigated by using different techniques and the results were correctly interpreted. A potential flame retardation mechanism was also presented.
Therefore, I recommend the article for publication in “Journal of Molecular Sciences”.
